# Atomistic Simulation of the Strain Driven Phase Transition in Pure Iron Thin Films Containing Twin Boundaries

**Yunqiang Jiang** [1], **Binjun Wang** [1,*], **Chun Xu** [1,*] **and Jianguo Zhang** [2]

1   School of Material Science and Engineering, Shanghai Institute of Technology, Haiquan Road 100, Shanghai 201418, China; 186082110@mail.sit.edu.cn

2   School of Mechanical Engineering, Shanghai Institute of Technology, Haiquan Road 100, Shanghai 201418, China; jgzhang98328@163.com

*   Correspondence: wangbj@sit.edu.cn (B.W.); xuchun1963@163.com (C.X.); Tel.: +86-21-60873525 (B.W.)

**Abstract:** Using molecular dynamics (MD) simulation, the strain-induced phase transitions in pure body-centered-cubic (bcc) iron (Fe) thin films containing twin boundaries (TBs) with different TB fractions and orientations are studied. Two groups of bcc thin films with different TB-surface orientation relationships are designed. In film group 1, the (112) [11$\bar{1}$] TBs are perpendicular to the (11$\bar{1}$) free surfaces, while the (112) [11$\bar{1}$] TBs are parallel to the free surfaces in film group 2. We vary the TB numbers inserted into the films to study the effect of TB fraction on the phase transition. Biaxial strains are applied to the films to induce the bcc to close packed (cp) phase transition. The critical strain, at which the first phase transition takes place, decreases with the TB fraction increase in film group 1 with a perpendicular TB-surface orientation, while such a relationship is not observed in film group 2 with parallel TB-surface orientation. We focus on the free surface and TB as the nucleation positions of the new phase and the afterward growth. In addition, the dynamics of the phase transition is discussed. This work may help to understand the mechanism of phase transition in nanoscale or surface-dominant systems with pre-existing defects.

**Keywords:** solid-solid phase transition; free surface; twin boundary; thin film; molecular dynamics simulation

## 1. Introduction

Solid-solid phase transition is an important and fundamental issue in the material science since it directly controls the microstructure of the materials. A typical example is the temperature changing induced $\alpha/\gamma$ transition in Fe, which has been intensively studied for many years [1,2]. In addition, there exists another allotropic transition in Fe, the $\alpha/\varepsilon$ transition [3–7]. Fe transforms from its original body-centered-cubic (bcc) structure to hexagonal close packed (hcp) structure at a pressure of around 13 GPa [8,9]. Studies about solid-solid phase transition in Fe were mostly carried out in bulk systems. In such systems, it is well known that defects normally serve as the nucleation positons because they provide the necessary energy and structure fluctuations for the new phase nucleation. In the early time, Caron et al. [10] found that dislocations associated with a group of twin boundaries (TBs) are the preferred sites for the martensite nucleation by experiment. Karewar et al. [11] investigated the effect of pre-existing TBs and stacking faults (SFs) in the parent face-centered cubic (fcc) phase on the martensitic phase transition and analyzed the atomic shear and transition pathways in presence of different defect structures. Recently, Luu et al. [12] determined the effects of twins, dislocations, and Cottrell atmospheres on the hydrostatic compression induced $\alpha \rightarrow \varepsilon$ transition in Fe by using the Monte Carlo methods and classical molecular dynamics simulation. The authors reported that the

existence of edge dislocation and TBs ease the hcp phase nucleation and concluded that the observed nucleation behavior differs from the results for pure Fe single crystals where the hcp phase nucleate randomly in the bulk material [13,14]. However, these works were carried out in bulk systems. Then, how is the situation in nanoscale or surface-dominant systems containing pre-existing defects, when a remarkable amount of free surfaces are in presence?

In this context, interest is increasing in studying phase transition in nanoscale or surface-dominant systems, such as nanoparticles and nanowires. Due to their small size caused extremely high surface-volume ration, nanoscale systems mostly exhibit dramatically changed properties compared with bulk systems, e.g., obviously enhanced tensile strength [15–17] or ductility [18]. Among nanoscale systems, thin films are two-dimensional nanomaterials whose properties are decisively influenced by the surface atoms. The mechanical properties of aluminum (Al) and tungsten (W) films on different substrates have been studied by Saha et al. using nanoindentation [19]. A cobalt (Co)-tantalum (Ta)-based metallic glass thin film with excellent fracture toughness and strength was designed by Kontis et al. [20]. Hardwick [21] reviewed the mechanical properties of free-standing thin films. Furthermore, solid-solid phase transition in thin films, especially in pure Fe thin films has received many attentions, since Fe is the basic material of steel. Using transmission electron microscope (TEM), Teodorescu et al. [22,23] studied laser induced bcc to face-centered cubic (fcc) phase transition in Fe thin films and found an incomplete transition with a fine mixture of fcc and bcc phases. Memmel et al. [24] observed a fcc (001) to bcc (110) transition during the growth process of Fe thin films on a Cu (001) substrate. Cuenya et al. [25] investigated the $\alpha/\gamma$ phase transition in ultrathin Fe films on $Cu_3Au$ (001) substrate and discussed the dependency of film thickness on the transition temperature. Up to now, it has been difficult to monitor the detailed phase transition process of Fe thin films by experiment. This is due to the reality that the non-diffusional phase transitions, for instance, the martensitic transition, proceed normally very fast. In this context, MD simulation is a powerful tool to investigate the solid-solid phase transition. By using this method, Urbassek and his coworkers studied both temperature [26] and strain [27,28] induced phase transitions in pure Fe thin films. A rich variety of phenomenon, such as transformation from a single crystal to a polycrystal and afterward grain refinement with strain increase, are reported. The phase transitions in these works mostly start at the free surface and the transition temperature/strain and pathway were often discussed in dependence of surface fraction (film thickness) and orientation. However, we notice that these simulations are performed in defect free single crystals. What about the situation when defects pre-exist in the thin film? Only a few studies are related to this issue. In the nanoscale system, Meiser and Urbassek [29] simulated the Fe thin films containing dislocations and reported the thermally induced phase transition is initiated at the dislocation positions in despite of the existence of free surface. Here we should note that dislocation is only one defect type in the material. It is too early to draw the conclusion that defects dominantly assist the phase transition rather than free surface.

We choose TB as the pre-existed defect in bcc Fe thin film to study the roles of defect and free surface in strain induced phase transition. The main reason for this choice is that TBs normally exhibit smaller energy compared to dislocations or grain boundary (GB). Since it has been confirmed that dislocations assist the phase transition in Fe thin films [29], it is meaningful to check whether another type of defect (TB), which is energetically more stable, has similar effect. In the present work, biaxial tensile strains will be applied to the bcc Fe thin films containing various TBs. We investigate the strain induced phase transition with respect to the TB fraction and orientation. The dynamics of the phase transition will be discussed.

## 2. Simulation Method

The interatomic potential plays a critical role in MD simulation. There are several potentials to describe the interaction between the Fe atoms, for instance, the Finnis–Sinclair (F-S) potential [30] or the Mendelev potential [31], which are widely used for different processes, such as fatigue behavior [32] or self-diffusion [33] in Fe. For $\alpha/\gamma$ phase transition, Engin et al. [34] analyzed six available interatomic

potentials for Fe and concluded that the Meyer–Entel potential [35] is the only one in the embedded atom method (EAM) class, which can describe both the austenitic and martensitic phase transitions. We choose the Meyer–Entel potential for our simulation. This is due to three reasons. Firstly, the Meyer–Entel potential provides relative high calculation speed due to its simple EAM form. Secondly, it has been affirmed that this potential can reproduce strain induced $\alpha/\varepsilon$ phase transition [27,28]. Lastly and most importantly, the Meyer–Entel potential gives correct energies of the free surfaces [26,36] and the planar defects [11], which are both involved in our simulation.

We design eight bcc Fe thin films with different number of TBs/without TBs to study the TB fraction and orientation dependency on the phase transition. In films 1–4, the free surfaces are orientated in the [11$\bar{1}$] directions and in films 5–8, the free surfaces are orientated in the [112] directions, which are the z-directions descripted in Table 1. The TBs contained in films 2–4 and 6–8 exhibit the same twinning structures, i.e., (112) twin planes with [11$\bar{1}$] twinning directions, which corresponds to the typical TB structure in bcc metals. According to this orientation relationship, the eight thin films can be divided in two groups. In group 1 with films 2–4, the (112) [11$\bar{1}$] TBs are perpendicular to the (11$\bar{1}$) free surfaces, while in film group 2 with films 6–8, the TBs are parallel to the (11$\bar{1}$) free surfaces. As a reference, films 1 and 5 do not contain any grain boundaries, i.e., they are single crystalline films. The detailed information of our simulated films is listed in Table 1.

**Table 1.** Detailed information of the simulated thin films. The coordinate directions are indicated by x, y and z (surface normal). $\Delta$x, $\Delta$y, and $\Delta$z are the thicknesses in each direction. T and N denote the number of TBs in the films and the total atom number, respectively.

| Film | x | y | z | $\Delta$x (Å) | $\Delta$y (Å) | $\Delta$z (Å) | T | N |
|------|-----|-----|-----|--------|--------|--------|---|---------|
| 1 | [1$\bar{1}$0] | [112] | [11$\bar{1}$] | 202.94 | 210.90 | 89.48 | 0 | 324,000 |
| 2 | [1$\bar{1}$0] | [112] | [11$\bar{1}$] | 202.94 | 210.90 | 89.48 | 2 | 324,000 |
| 3 | [1$\bar{1}$0] | [112] | [11$\bar{1}$] | 202.94 | 210.90 | 89.48 | 4 | 324,000 |
| 4 | [1$\bar{1}$0] | [112] | [11$\bar{1}$] | 202.94 | 210.90 | 89.48 | 6 | 324,000 |
| 5 | [1$\bar{1}$0] | [11$\bar{1}$] | [112] | 215.12 | 211.27 | 84.36 | 0 | 324,360 |
| 6 | [1$\bar{1}$0] | [11$\bar{1}$] | [112] | 215.12 | 211.27 | 84.36 | 1 | 324,360 |
| 7 | [1$\bar{1}$0] | [11$\bar{1}$] | [112] | 215.12 | 211.27 | 84.36 | 2 | 324,360 |
| 8 | [1$\bar{1}$0] | [11$\bar{1}$] | [112] | 215.12 | 211.27 | 84.36 | 3 | 324,360 |

Figure 1 shows films 2 and 7 as examples for the simulation setup of film groups 1 and 2. We choose film 2 as an example to clarify the modelling details. Based on the lattice constant of bcc Fe, which is predicted as 2.87 Å by the Meyer–Entel potential, a simulation box with the dimensions of 202.94 × 70.30 × 89.48 Å with the orientations listed in Table 1 is constructed. Note that the 70.30 Å in the y direction corresponds exactly 1/3 of the film 2. This simulation box is filled with Fe atoms in bcc structure. Then, a mirror symmetry operation is performed to the constructed crystal to produce a mirrored crystal. The mirror plane is normal to the y direction ([112]) and placed at y = 0. The mirrored crystal is positioned directly on the left side of the original one. Now the system has a dimension of 202.94 × 140.60 × 89.48 Å and contains a (112) [11$\bar{1}$] TB in the middle. Afterwards, the mirrored crystal is duplicated and placed on the right side of the original crystal. Film 2 with two (112) [11$\bar{1}$] TBs is hereto constructed, see Figure 1a. Films 3 and 4 with more TBs are constructed in analogous way by alternate placement of the original crystal and the mirrored crystal. For films 6–8 with the parallel orientation, the mirror planes, whose plane normal correspond to the z direction ([112]), are placed at z = 0. The mirrored crystal(s) is (are) stacked on the top of the original one, see Figure 1c as an example.



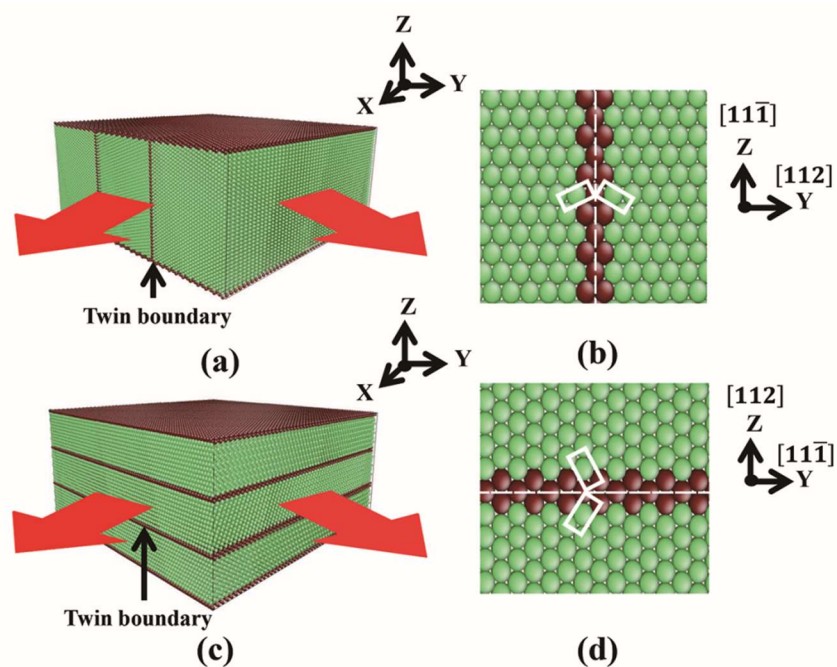

**Figure 1.** Simulation setup. Colors denote the local crystal structure: green, bcc; brown, unknown. (**a**) Film 2 as an example for film group 1, where the TBs are perpendicular to the free surfaces; (**b**) Constructed $(112)_{bcc}$ TB in film 2; (**c**) Film 7 as an example for film group 2, where the TBs are parallel to the free surfaces; (**d**) Constructed $(112)_{bcc}$ TB in film 8. In b and d, the white dashed lines indicate the TBs and the white rectangles display the mirrored unit cells.

For all the films, periodic boundary conditions are employed in the x and y directions. The z direction is set as free due to the free surface. The constructed films are relaxed in an NPT ensemble for 50 ps. The temperature is controlled at 500 K by a Nosè–Hoover thermostat. Here we should note that the Meyer–Entel potential predicts an $\alpha/\gamma$ transition temperature of 550 K [34]. At this temperature, the bcc phase is still stable. It has been proved that the phase transition will be inhibited at low temperatures [27]. The pressure is controlled to zero in x and y directions. In z direction, no pressure control is performed.

After the relaxation, the simulations are performed in an NVT ensemble at a temperature of 500 K. Biaxial strains with the strain rate of $5 \times 10^8$ s$^{-1}$ are applied to the thin films along the x and y directions by the following way. After each simulation period of 1 ps, the whole simulation box is forced to expand 0.05% in the x and y directions. This means, after each straining, the film has 1 ps time to relax. The normal stresses $\sigma_{xx}$, $\sigma_{yy}$, and $\sigma_{zz}$ in x, y, and z directions for each straining are calculated as the average value of the monitored stresses at each time step (0.1 ps) during the 1 ps relaxation time. The hydrostatic stress $\sigma$ is given as:

$$\sigma = \frac{\sigma_{xx} + \sigma_{yy} + \sigma_{zz}}{3} \tag{1}$$

The stress-strain curves for all of the films are plotted. We also monitor the critical strains, at which the phase transitions take place and capture the representative snapshots during the process occurring.

The films are constructed by using the program ATOMSK (Beta 0.10.6; The University of Lille, Lille, France) [37]. All of the simulations are carried out with the open source MD simulation code LAMMPS [38]. The dislocation extraction algorithm (DXA) [39] is used to identify the possible dislocation formation during the biaxial straining. We use common neighbor analysis (CNA) [40] to identify the local lattice structure. The visualization is realized by using the Atomeye [41] code.

## 3. Result

After the relaxation, the TBs remain their original crystalline structures and positions in film 1–7. This is not unexpected because most of the Fe potentials can reproduce stable TBs both in bcc and fcc structure [42,43]. Only in film 8, we observe TB migration during the relaxation. The upper and lower TBs migrate in the direction of the free surfaces, see Figure 2. Here we should note that the TB spacing of film 8 is only 2.1 nm. The repulsive force offered by the TBs [43] causes this twin migration. We check the possible dislocation initiation during the TB migration by DXA and no dislocations are found. Note that the relaxation is a pure thermal treatment and no external stresses/strains are applied to the system to induce the dislocation nucleation. Thus, we may assume that this TB migration would not influence the phase transition behavior. Finally, note that pressure in the z directions of the films are relaxed dynamically to zero due to the free surfaces.

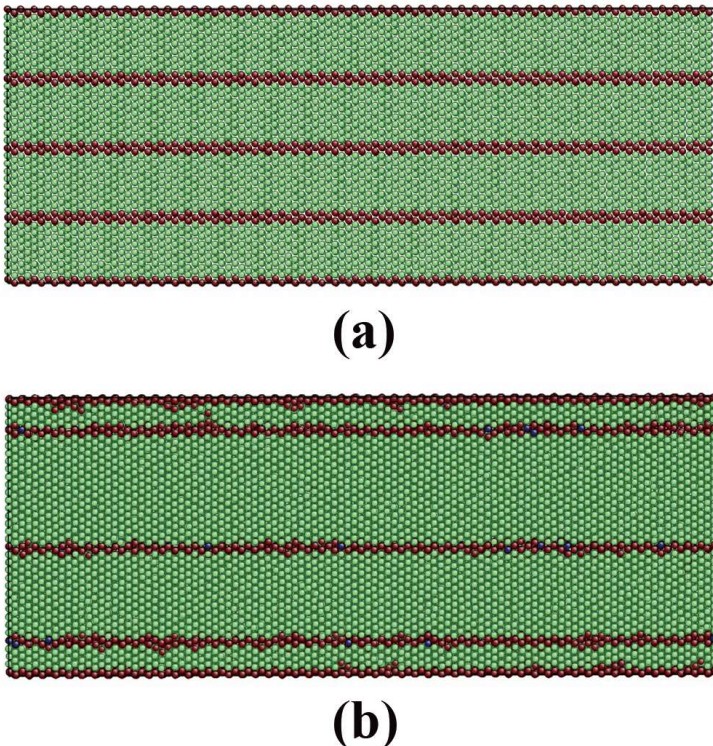

**Figure 2.** Migration of TBs during relaxation in film 8. Colors denote the local crystal structure as in Figure 1. (**a**) The (111)$_{bcc}$ plane before the relaxation; (**b**) The (111)$_{bcc}$ plane after the relaxation.

### 3.1. Film Group 1 with Perpendicular TB-Surface Orientation Relationship

3.1.1. Stress-Strain Curves

Figure 3 shows the stress-strain curves of films 1–4. All the films exhibit a linear behavior at the beginning, which corresponds to an elastic deformation in the original bcc films. TB fraction does not have influence on elastic modulus. Sainath and Choudhary [43] simulated uniaxial tensiled bcc nanopillars containing TBs and obtained the same result. The tensile strength of the films decreases with TB spacing decrease, i.e., reverse Hall–Petch effect [44,45], which is often observed in nanoscale polycrystals [46–48]. Xu and He [49] investigated the mechanical properties of bcc Fe nanowires containing TBs with different TB spacing using MD simulation with the Ackland interatomic potential [50]. This reverse Hall–Petch effect was also reported by the authors. At the end of the elastic regimes, the stresses are released dramatically. We check our simulated films 1–4 after the elastic regimes by DXA. No dislocations can be detected and this indicates that the high stresses are relaxed by another process, in our case the phase transition, rather than the dislocation motion. Note that

for films 1–4, the strain, at which the first phase transition occurs, corresponds roughly to the strain, at which the stress dramatically decreases. The detailed process occurring will be discussed later in Section 3.1.2.

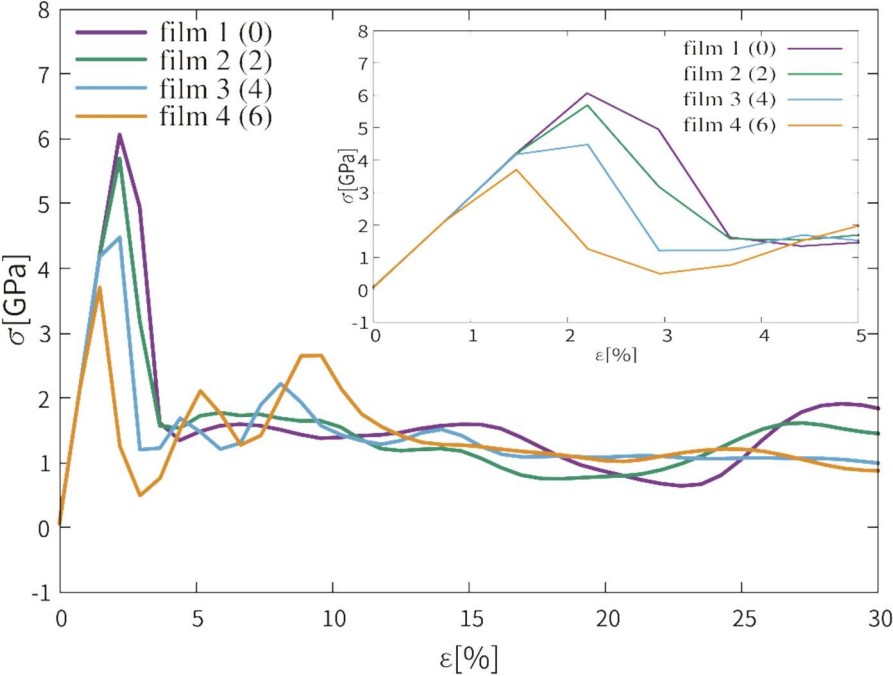

**Figure 3.** Stress-strain curves of films 1–4. The numbers in brackets indicate the numbers of TB in the films. The subfigure inserted shows the zoomed stress-strain curves in the strain range of 0–5%.

### 3.1.2. Phase Transitions in Films 1–4

Figure 4 shows the evolution of the fractional phase content as a function of the applied strain. The first phase transition takes place at the strains of 2.9%, 2.6%, 2.3%, and 2.0% for films 1–4, respectively. These strains are in accordance with the ends of elastic regimes shown in Figure 3. These values show the tendency that the critical strain, at which the first phase transition occurs, decreases with TB fraction increase.

Now, we discuss the films separately. Firstly, we should note that the free surfaces in films 1–4 are orientated in $[11\bar{1}]_{bcc}$ direction. Using the same potential, Meiser and Urbassek [26] simulated Fe thin slabs with $(111)_{bcc}$ surfaces and they reported that no phase transition takes place during a heating range from 10 K to 2000 K. The authors regarded this free surface as an obstacle to phase transition in spite of its high surface energy of 1.76 J/m$^2$.

Figure 5 shows the representative snapshots of the local atomic structure of film 1 at different strains. The first nucleation of the new phase starts at the free surface, as always observed in thin film systems [26–28], see Figure 5a. Although the $(11\bar{1})_{bcc}$ surface is not conserved in any common phase transition pathways [26], for instance, the Bain pathway [51], the Kurdjumov–Sachs (K–S) relationship [52] or the Nishiyama–Wassermann (N–W) relationship [53], the distance between the atoms becomes larger due to the applied strains in x and y directions. This may provide more free volume for the coordinate movement of the atoms, which benefits the new phase nucleation. Then, from the free surface nucleated new phases grow into the film, while homogenous nucleation in the interior of the film also takes place, as indicated by the white rectangle in Figure 5b. The new phase mainly consists of hcp phase, but fcc grains in stripe form can be also observed, see Figure 5c. Normally, stress induces a bcc → hcp phase transition. The existence of the fcc phase here is due to two reasons. Firstly, the free energy difference between the fcc and hcp phases is tiny, which amounts to 4 meV/atom [54]. Small temperature or pressure fluctuations during the simulation can induce

transition between the two phases. Secondly, the fcc and hcp phase differ only in stacking sequence. The applied strains act exactly as the shear to shift fcc (hcp) atom layers to generate hcp (fcc) phase. Thus, we may denote this phase transition as a bcc → close packed (cp) transition rather than bcc → fcc or bcc → hcp transition. Increasing the strain, the phase transition deaccelerates and at a strain of 12%, the first phase transition is almost finished, as shown in Figure 5c. The cp phases nucleated from the surface exhibit different crystalline orientation as the homogeneously nucleated ones. Thus, a polycrystalline thin film is formed after the transition. At strains around 20% and 32%, we observe two partial back transitions from the cp phase to the bcc phase, see Figure 4a. Figure 5d displays the first one and the second one is similar. In previous works about phase transition in nanowires [55] or thin films [27], such back transitions have been also reported. The authors contributed this to the applied strains, which act as the necessary shear strain for the martensitic phase transition. The back transformed bcc phase has different orientation as the initial one. It is unstable and transforms to cp phase with strain increase. During the whole process occurring, the hcp and fcc phase transform to each other and the cp grains are reoriented to the favorable orientations to the applied strains by GB migration, c.f., Figure 5c,d. We conclude that both homogenous (in bulk) and inhomogeneous (at the free surface) nucleation are involved in the phase transition.

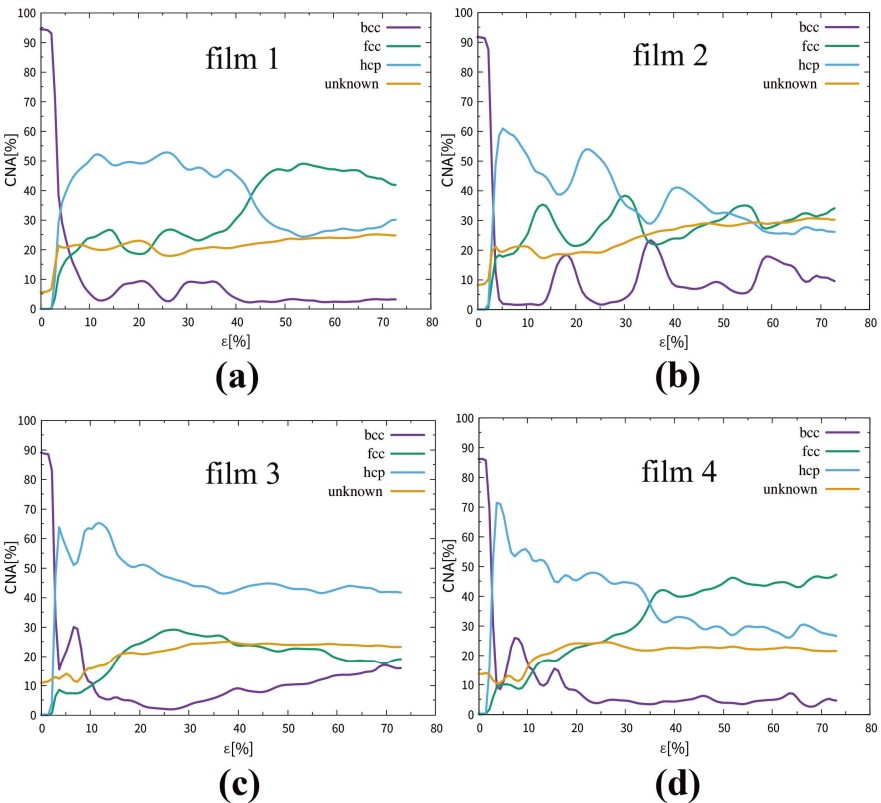

**Figure 4.** Fractional phase content as a function of the applied strain ε of films 1–4. (**a**) Single crystal film 1; (**b**) Film 2 with two $(112)_{bcc}$ TB; (**c**) Film 3 with four $(112)_{bcc}$ TBs; (**d**) Film 4 with six $(112)_{bcc}$ TBs. TBs in these films are perpendicular to the free surfaces.

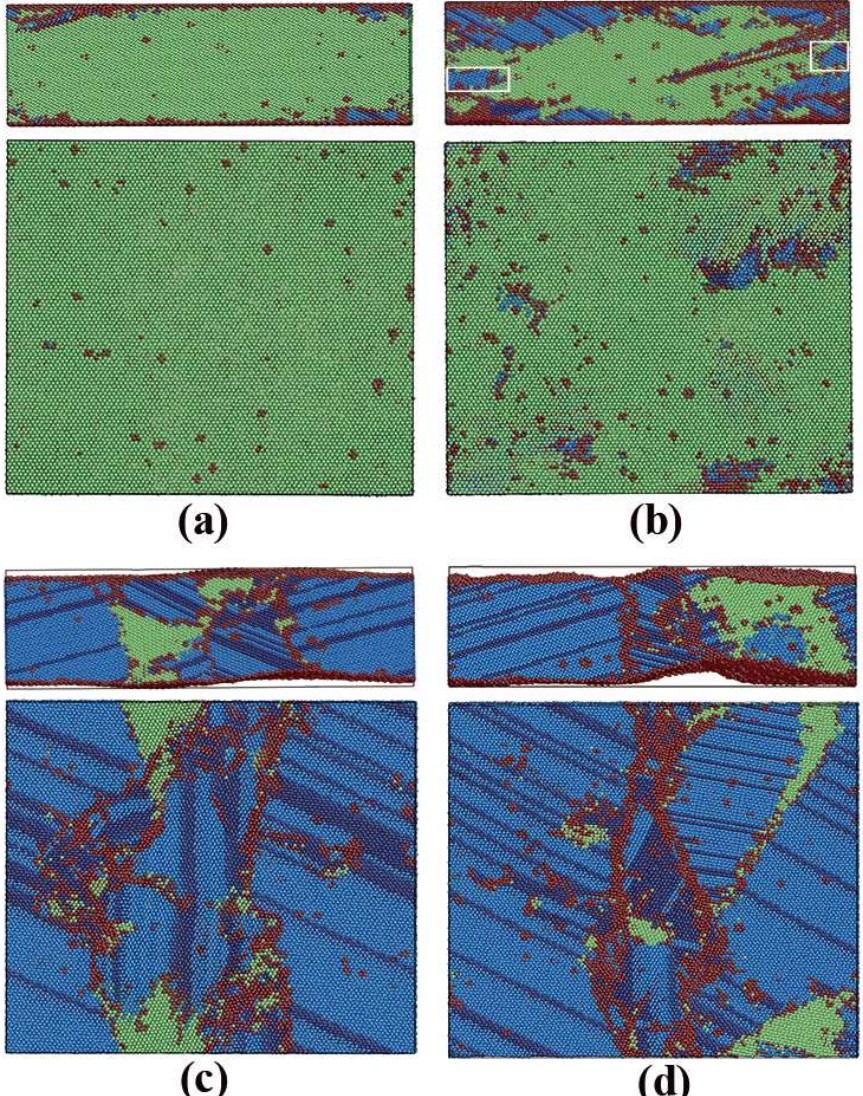

**Figure 5.** Snapshots of the local crystal structure at different strains of film 1. Colors denote the local crystal structure: green, bcc; dark blue, fcc; light blue, hcp; brown, unknown. In a–d, the upper figures show the original $(1\bar{1}0)_{bcc}$ plane and the lower figures display the original $(11\bar{1})_{bcc}$ planes. For the lower figures, the film is cut along the $(11\bar{1})_{bcc}$ plane in the middle and the upper part is removed. (**a**) Local atomic structure at a strain of 2.8%. The first nucleation starts at the free surface; (**b**) Status at a strain of 5.6%. The white rectangles show the homogeneously nucleated cp phases in the bulk material; (**c**) Status at a strain of 12%. The first phase transition is almost finished; (**d**) Partial back transition to the bcc phase at a strain of 20%.

Phase transition in film 2 containing two TBs is more complicated. Figure 6 shows the snapshots of the ongoing phase transition. The first nucleation takes place at the position, where the TB and the free surface intersect, as shown in Figure 6a. This is not unexpected because this area offers the largest energy and structure fluctuations, which are necessary for the new phase nucleation, in the whole system. Karewar et al. [11] reported the convergences of TBs or TB-SF are the favorable martensitic nucleation positions, which is similar to our observation. The new cp phases grow in the diagonal directions, as shown in the upper portion of Figure 6b, while the new cp phase also nucleates at the TB, see the lower portion of Figure 6b. Interestingly, surface nucleation can be also observed with the strain increase, which is indicated by the white rectangle in Figure 6c. Here we should note that the corresponding strains of Figure 6a–c differ only 0.2%. Thus, we may consider simultaneous nucleation at different positions. In Figure 6d, the three types of cp phase, which nucleated at the intersection

area between TB and the free surface, free surface and TB, coexist. The three cp phases are indicated by the numbers 1, 2, and 3 in the upper portion of Figure 6d. The type 3 cp grains have the most favorable orientation to the applied strains and they "devour" part of the type 1 and 2 grains as the strain increases, as shown in Figure 6e. The original TBs do not lose their mirrored structure and exist as new TBs in the cp phase (see red lines in Figure 6e). Finally, two similar back transitions to the bcc phase are also observed at high strains (see Figure 4b). Figure 6f shows the first one. In this case, we conclude that TB, the free surface, and their intersection area act as the nucleation positions for the new phase.

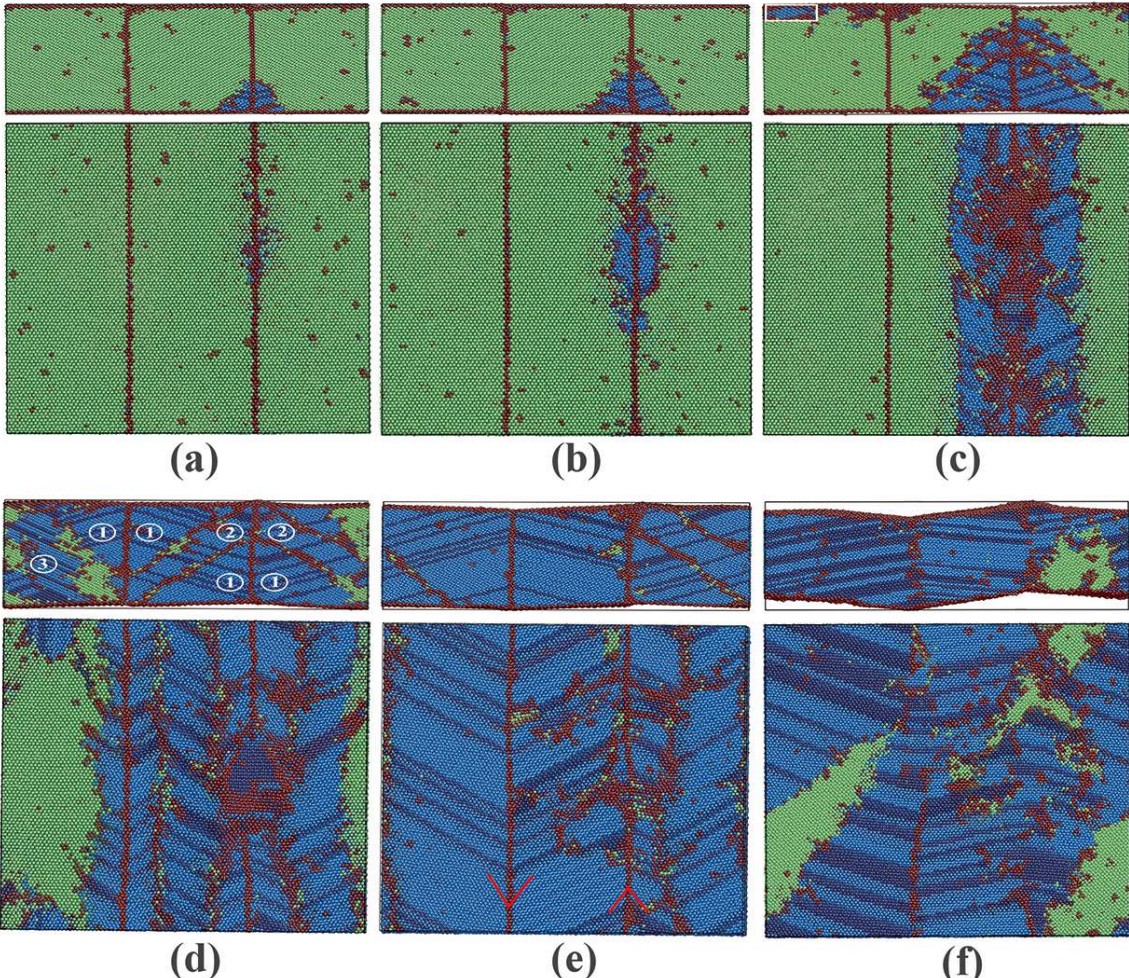

**Figure 6.** Snapshots of the local crystal structure at different strains of film 2. Colors denote the local crystal structure as in Figure 5. In a–f, the upper and lower figures show the same crystalline plane as in Figure 5. (**a**) Local atomic structure at a strain of 2.6%. The first nucleation starts at the intersection area between the free surface and TB; (**b**) Status at a strain of 2.7%. The new phase also nucleates at the TB; (**c**) Status at a strain of 2.8%. Growth of the nucleated cp phase and new nucleation at the free surface is indicated by the white rectangle; (**d**) Status at a strain of 3.2%. The numbers 1, 2, and 3 indicate the new phase nucleated from different sources, for details see context; (**e**) Status at a strain of 4.5%. The formed twin structure in the hcp phase is denoted by the red lines; (**f**) Partial back transition to the bcc phase at a strain of 18%.

For film 3, we observe a TB tilting process before the phase transition (see the upper portion of Figure 7a). No dislocations are found during this TB tilting by using DXA. In film 3, the TB spacing is only around 5 nm. Thus, the repulsive forces [43] between the TBs are larger than that in film 2. Note that the strains, at which the TB tilting takes place, are still the elastic regime before the yield point is

reached. It is well known that the dislocations can be only nucleated in the plastic regime. Assuming the TB as a special form of GB, it has been often observed that grains are reoriented to the favorable orientations to the applied load [27,28]. We may regard this TB tilting as a result of reorientation of the grains induced by the applied strains and the repulsive forces between the TBs. The free surfaces exhibit a kink structure. This indicates that the film attempts to relax the high stress via the free surfaces rather than dislocations. The first nucleation takes place at the TBs (or in the intersection area between TB and the free surface) instead of the free surface. The nucleated cp (mainly hcp) phases grow rapidly in the transverse directions, c.f., Figure 7b,c. The fcc phase in the lower portion of Figure 7b consists of only one fcc atom layer and transforms to the hcp phase as the strain increases. The bcc phases in Figure 7c are very stable and do not disappear until the back transition takes place at a strain around 3% (not shown in the snapshots), see the bcc curve in Figure 4c. After the first bcc → hcp phase transition, we obtain an almost single crystalline hcp film, as shown in Figure 7c. The fcc phases in stripe form consist mostly of one or two atom layers, which are regarded as stacking faults in the hcp structure. Figure 7d shows the snapshot of the film at a very high strain of 24%. This polycrystalline structure is mainly from three different sources. (i) The original transformed single crystalline hcp phase; (ii) the mutual transition between hcp and fcc phases at high strains; and (iii) the reorientation of the grains whose orientations are unfavorable to the applied strains. In addition, the free surfaces become very rough high strains, as shown in the upper portion of Figure 7d. We also monitor the possible dislocation generation during the whole process and no dislocation can be found. Thus, this rough surface shows that the phase transition or reorientation-induced stresses are relaxed by the free surface.

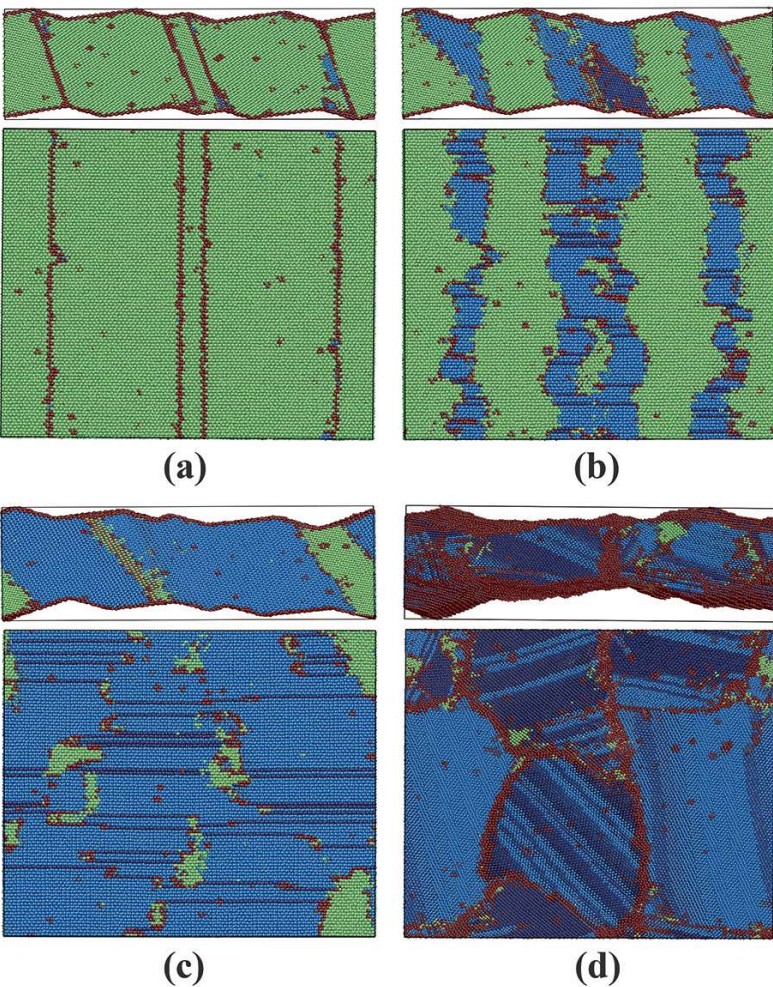

(a)  (b)  (c)  (d)

**Figure 7.** Snapshots of the local crystal structure at different strains of film 3. Colors denote the local crystal structure as in Figure 5. In a–d, the upper and lower figures show the same crystalline plane as in Figure 5. (**a**) Local atomic structure at a strain of 2.3%. The first nuclei appear at the TB; (**b**) status at a strain of 2.6%. Growth of the hcp phases nucleated from the TBs; (**c**) status at a strain of 3.0%. An almost single crystalline hcp film is formed; and (**d**) a polycrystalline cp film at a strain of 24%.

The process occurring of film 4 is similar to that of film 3. The whole process can be described as five steps as strain increases: (i) TB titling; (ii) nucleation at the TBs; (iii) growth of the hcp phases and formation of a single crystalline film; (iv) partial back transition to the unstable bcc phase; and (v) transition to a polycrystalline cp film. Here we should note that differing from film 2, the main nucleation of the new cp phases in film 3 and 4 start at the TBs, no nucleation in the bulk material or at the free surfaces is observed.

### 3.2. Film Group 2 with Parallel TB-Surface Orientation Relationship

### 3.2.1. Stress and Strain Curves

Figure 8 shows the stress-strain curves of films 5–8. The curves exhibit similar tendency in the strain range from 0% to 10%. This is no wonder since no normal strain is applied to the TBs, which are parallel to the free surfaces. It has been proved that twin spacing does not influence the mechanical behavior of gold (Au) nanowire, when the TBs are parallel to the tensile direction [56]. For all the films 5–8, the stresses drop dramatically after the first elastic regime. All the curves exhibit a second and a third elastic regime. Analogous to films 1–4, the strains, at which the stresses dramatically drop, correspond roughly to the strains, at which the phase transitions or violent microstructure evolutions take place.

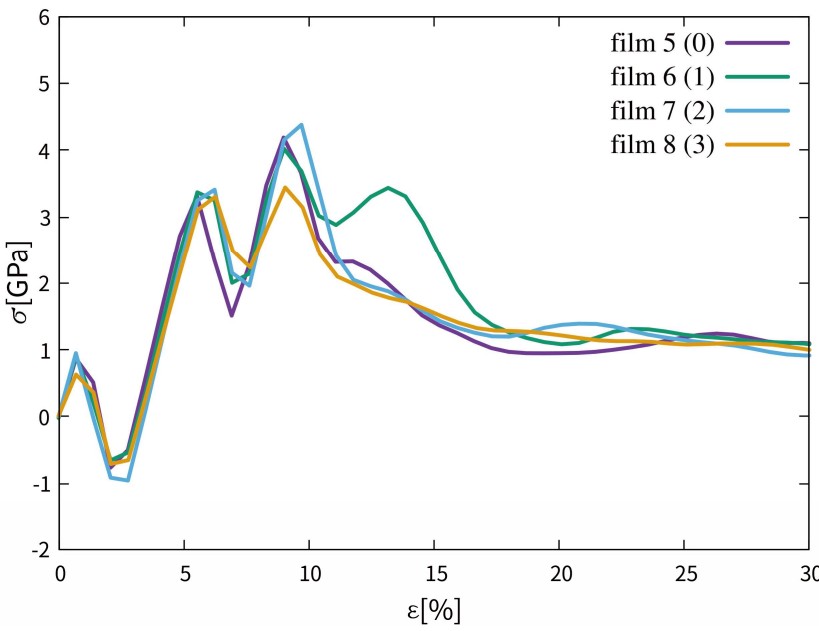

**Figure 8.** Stress-strain curves of films 5–8. The numbers in brackets indicate the numbers of TB in the films. The subfigure inserted shows the zoomed stress-strain curves in the strain range of 0–5%.

### 3.2.2. Phase Transitions in Films 5–8

Figure 9 displays the evolution of the fractional phase content as a function of the applied strain of films 5–8. The bcc and hcp curves show very similar tendency for all of the films. The bcc fractions drop to almost zero within the strain of 3%, while the hcp fractions increase to a value over 90%. This indicates complete bcc → hcp phase transitions. The back transformations to bcc phase and its

further transition to cp phase can be observed within the strain of 15%. At high strains, the fractions of the hcp phase decrease, while that of the fcc phase increase. Although this similarity, we should check the phase transition and microstructure evolution for each film in detail.

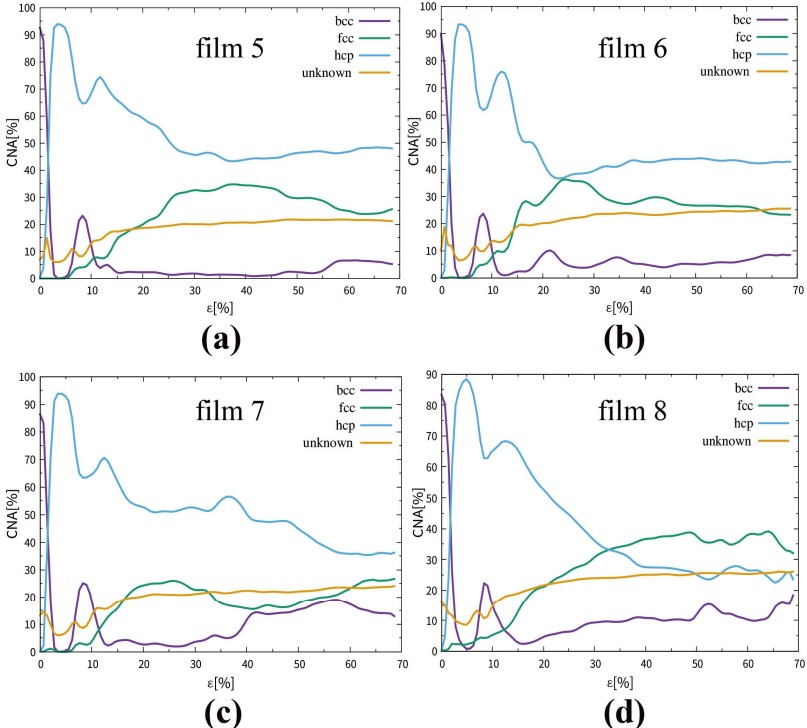

**Figure 9.** Fractional phase content as a function of the applied strain of film 5–8. (**a**) Single crystalline film 5; (**b**) Film 6 with one $(112)_{bcc}$ TB; (**c**) Film 7 with two $(112)_{bcc}$ TBs; (**d**) Film 8 with three $(112)_{bcc}$ TBs. TBs in these films are parallel to the free surfaces.

Figure 10 shows the microstructure evolution of film 5, which is a single crystal with $[112]_{bcc}$ oriented free surfaces. We calculate the $(112)_{bcc}$ surface energy using the Meyer–Entel potential and obtain a value of 3.34 J/m$^2$. This value is approximately doubly as high as that of the $(111)_{bcc}$, $(110)_{bcc}$, and $(100)_{bcc}$ surfaces, which are 1.76, 1.35, and 1.73 J/m$^2$, respectively [26]. This high surface energy may provide the necessary energy fluctuation for the new phase nucleation. As expected, the first nucleation starts at the free surfaces, as shown in Figure 10a. Differing from the surface nucleation in film 2, the new hcp nuclei cover the whole free surfaces rather than a certain position. This should be attributed to the high surface energy, which makes the nucleation possibility at each position at the free surface extremely high. The nucleated hcp phases grow rapidly into the film, as shown in Figure 10b. The hcp phases from the upper and lower surfaces have the same crystalline orientation and they combine to a single crystalline hcp film (see Figure 10c). Up to a strain of 6%, the single crystalline hcp film remains stable. This pure hcp film corresponds to the second elastic regime on the stress-strain curve of film 5 in Figure 8. Then, a partial back transition to bcc phase is observed and the maximal back transformed bcc phase fraction amounts to over 20% at a strain around 8%, as shown in Figure 10d. This bcc phase is unstable and transforms to cp phase with strain increase, not shown in the snapshots. The possible reason for this back transition has been discussed in Section 3.1.2. The mixture consisting of hcp and bcc phase is responsible for the third elastic regime on the stress-strain curve of film 5 in Figure 8. Finally, the single crystal hcp film transforms to a polycrystalline cp film with further strain increase, similar to the film 3 case, also not shown in the snapshots.

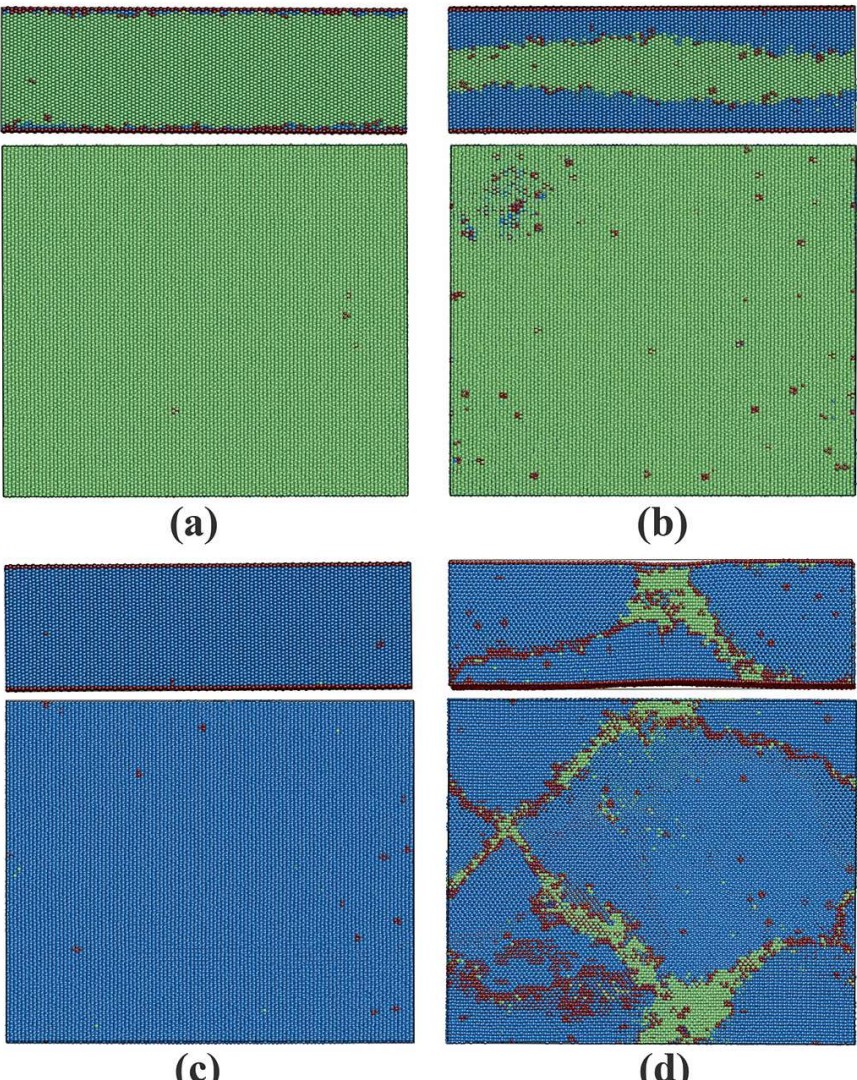

**Figure 10.** Snapshots of the local crystal structure at different strains of film 5. Colors denote the local crystal structure as in Figure 5. In a–d, the upper and lower figures show the same crystalline plane as in Figure 5. (**a**) Local atomic structure at a strain of 1%. The first nuclei cover the whole free surfaces; (**b**) status at a strain of 1.5%. The hcp phases grow into the film; (**c**) status at a strain of 2.9%. A single crystalline hcp film is formed; and (**d**) partial back transition to bcc phase at a strain of 6.2%.

Now the question is standing out. Do the TBs also act as the nucleation position, in despite of the existence of free surface with high surface energy? Films 6–8 exhibit similar phase transition behavior and film 7 is taken as an example to clarify the question. Figure 11 shows several representative snapshots of the first phase transition of film 7. It can be clearly observed that the new hcp phases nucleate both at the free surface and the TB simultaneously, as shown in Figure 11a. Papon et al. [57] calculated the $\{112\}_{bcc}$ TB energy and obtained a value of 0.171 J/m$^2$. In addition, the $\{112\}_{bcc}$ TB energy given by Shibuta et al. [42] is 0.6 J/m$^2$. These values are much lower than our calculated $\{112\}_{bcc}$ surface energy. It seems that the TBs are not the energetically favorable positions for the nucleation. However, another factor for the nucleation is the structure fluctuation in the parent phase. The $\{112\}_{bcc}$ TB consists of a stacking disorder of six atom layers with the ABCDEFAFEDCBA configuration [58]. Comparing the $\{111\}_{fcc}$ TB, which is caused by a simple stacking disorder as ABCACBA, we may assume $(112)_{bcc}$ TBs with big disorders may provide the necessary structure fluctuation for the nucleation. Once the nuclei are formed, they grow rapidly in the (opposite) surface normal directions until a perfect hcp film is obtained, as shown in Figure 11b–d. Once the growing hcp phase comes into contact with the

TB, at which no nucleation takes place, the TB is "devoured", c.f., the upper Figure 11c,d. We do not observe any halt during this "devouring" process. It can be concluded that the TBs serve as nucleation positions and will not be the obstacles of the afterward growing process. With further strain increase, the single crystal hcp film undergoes the partial back transition to bcc phase and transforms from a single crystalline hcp film to a polycrystalline cp film, analog to the descriptions in other films, detailed description see film 3 in Section 3.1.2.

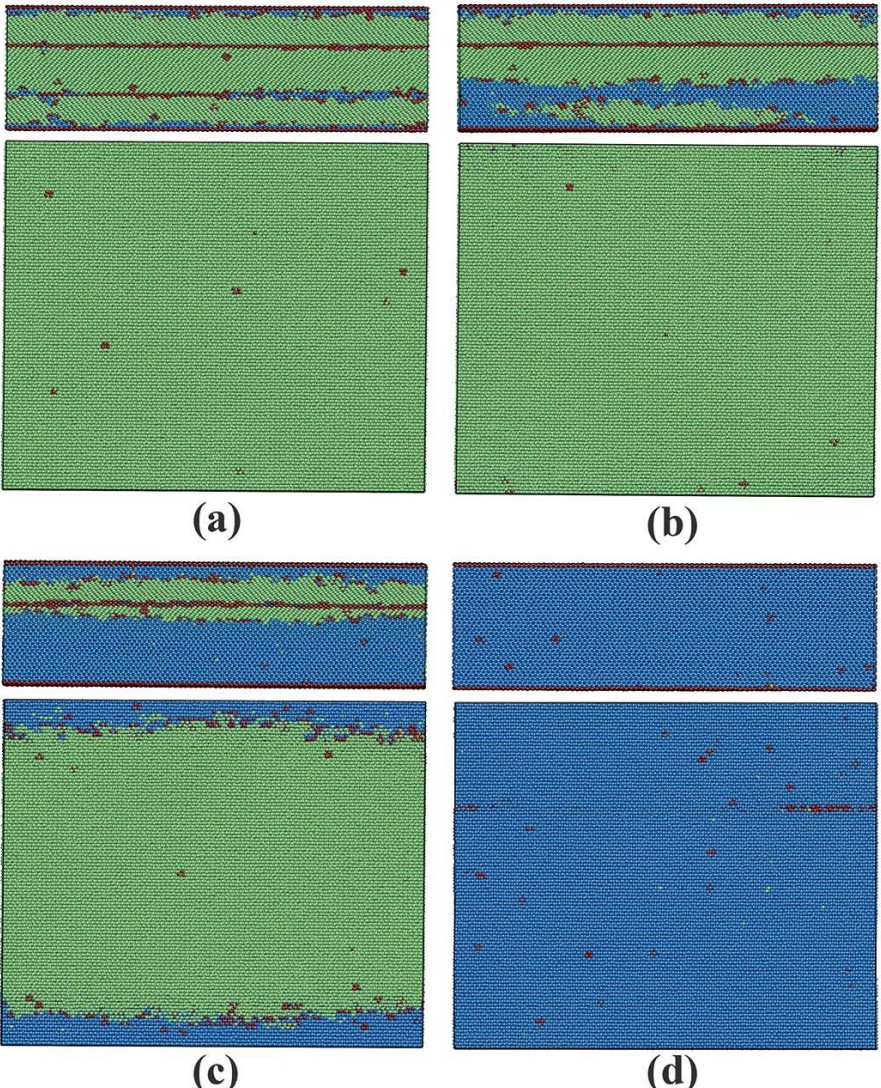

**Figure 11.** Snapshots of the local crystal structure at different strains of film 7. Colors denote the local crystal structure as in Figure 5. In a–d, the upper and lower figures show the same crystalline plane as in Figure 5. (**a**) Local atomic structure at a strain of 1%. Both the free surface and TB act as the nucleation positions; (**b**) status at the strain of 1.1%. The hcp phases grow into the film; (**c**) status at the strain of 1.2%. Further growth of the hcp phases; (**d**) status at a strain of 3%. A single crystalline hcp film is formed.

*3.3. Dynamics*

Now we analyze the dynamics of the first bcc → cp phase transition in the films. Figure 12 shows the strains, at which a particular transition degree is achieved, in dependency of the TB number in the films. In film group 1, where the TBs are perpendicular to the free surface, the curves are almost parallel and have a downward trend. This means firstly, the phase transition starts at a lower strain in films containing more TBs than films containing less TBs. This is logical because more TBs provide

more nucleation positions for the new phase. Secondly, the growth velocities of the new phase in all the films are roughly equivalent (except film 1 as a single crystal film at strains above 3.8%), whether the free surface attends to the nucleation (film 2). Here we should note that for film 1 as a single crystal, a transition degree of 80% is achieved at a high strain of 6%. This should be explained by the homogeneous nucleation, see Figure 5b, which deaccelerates the transition. For film group 2, where the TBs are parallel to the free surface, the curves of the films are parallel to each other. This means equivalent growth velocities of the hcp phase in all of the films. However, the downward trend with TB number increase is not observed. The critical strain, at which the phase transition takes place, does not show dependency on the TB fraction. We notice that the first nucleation occurs very early, i.e., at a very low strain of around 0.2%. This is due to the reality that both the $(112)_{bcc}$ surface with very high surface energy and $(112)_{bcc}$ TB with large scale of disorder provide high possibilities for the nucleation. Even when the TBs would not work, the free surface is able to offer enough nucleation positions for the new phase, see the "devoured" TB in Figure 11c.

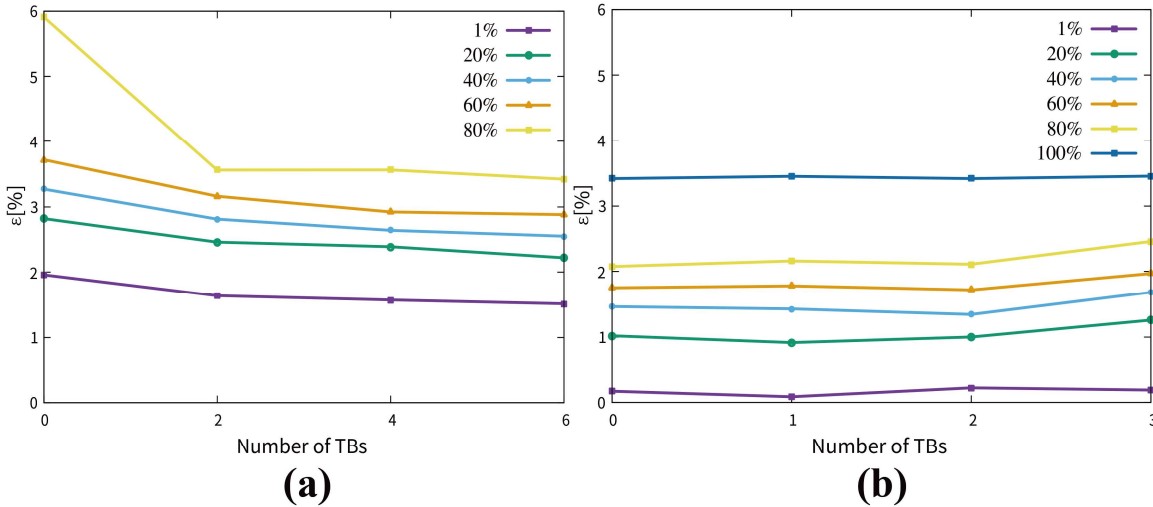

**Figure 12.** Dependence of the strain, at which the fractions of the cp phase indicated in the legend have been achieved of the first bcc → cp transition, on the number of TBs in the films. (**a**) Film group 1, where the TBs are perpendicular to the free surfaces. The subfigure inserted indicates the zoomed strain range from 1% to 4%; and (**b**) Film group 2, where the TBs are parallel to the free surfaces.

## 4. Conclusions

Using MD simulation, the strain induced phase transitions in Fe thin films containing TBs with different fraction and orientation have been investigated. The main proposal of this work is to study the roles of free surface and TB on the phase transition. The whole transition processing for all the films simulated can be roughly divided into five steps. (i) Nucleation at certain positions; (ii) growth of the new phases; (iii) partial back transitions; (iv) transition of the back transformed bcc phase; and (v) refinement of the grains (films 1 and 2) or transition from a single crystalline film to a polycrystalline film (films 3–8).

In film group 1, where the $(112)_{bcc}$ TBs are perpendicular to the $(11\bar{1})_{bcc}$ free surfaces, the films show reverse Hall–Petch relationship. For film 1 with the $(11\bar{1})_{bcc}$ free surface, both homogeneous nucleation in the bulk material and heterogeneous nucleation at the free surfaces contribute to the phase transition. The homogeneous nucleation is the reason for the deceleration of the phase transition at high strains. For film 2 containing two TBs, the new cp phase originate from three sources, namely, the free surfaces, TBs and the intersection area between them. Increasing the TB fraction (films 3 and 4), the new phases nucleate at the TBs rather than the free surface. The critical strain, at which the first phase transition takes places, decreases with the TB fraction increase.

In film group 2, where the $(112)_{bcc}$ TBs are parallel to the free surfaces, both the free surface and the TB assist the phase transitions as nucleation positions (films 6–8). Dependence of TB fraction on the critical strain, at which the first phase transition takes place, is not found. The reason for this is the high nucleation possibilities at the free surfaces and TBs, which are caused by the high surface energy and the large scale disorder, and the rapid growth afterward. In addition, only the free surfaces serve as the nucleation positions in the single crystalline film 5. The first nuclei cover the whole free surfaces and grow into the film.

Comparing film groups 1 and 2, the main differences between the transition behaviors can be concluded as follows. Firstly, the TB fraction influents the nucleation mechanism in film group 1 with the perpendicular TB-surface orientation relationship. The effect of the free surface as nucleation positions vanishes with TB fraction increase. In film group 2 with the parallel TB-surface orientation relationship, this influence of TB fraction on the nucleation mechanism is not found. All the films in this group exhibit analogous transition behavior. Secondly, the critical strain, at which the first phase transition takes place, decreases with TB fraction increase in film group 1, while this dependence is not observed in film group 2. Lastly, the strain ranges, in which the phase transitions complete, are generally lower in film group 2 than that in film group 1. This is due to the active $(112)_{bcc}$ surface with high surface energy. This work may help to understand the phase transition mechanism in nanoscale systems containing pre-exist TBs.

**Author Contributions:** Y.J. and J.Z. performed the MD simulations and analyzed the results. B.W. and C.X. designed the work. All the authors discussed the results and Y.J. and B.W. wrote the manuscript. All authors have read and agreed to the published version of the manuscript.

**Funding:** This research was funded by Shanghai Pujiang Program, grant number 17PJ1408600 and Collaborative Innovation Program of Shanghai Institute of Technology, grant number XTCX2019-8.

**Acknowledgments:** J. G. Z acknowledges the support by Shanghai Natural Science Foundation of China (grant no: 19ZR1455100).

**Conflicts of Interest:** The authors declare no conflict of interest.

## Nomenclature

| | |
|---|---|
| x, y, z | Coordinate directions |
| $\Delta x$, $\Delta y$, $\Delta z$ | Thicknesses of the films in each direction, Å |
| T | Number of the TBs in the films |
| N | Total number of the atoms |
| $\sigma$ | Hydrostatic stress, GPa |
| **$\sigma_{xx}$, $\sigma_{yy}$, $\sigma_{zz}$** | Normal stresses in x, y and z directions, GPa |
| $\varepsilon$ | Strain, % |
| CNA (in **Figure 4** and **Figure 9**) | Fractional phase content, analyzed using common neighbor analysis (CNA), % |

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
