# Peer review of "Atomistic Simulation of the Strain Driven Phase Transition in Pure Iron Thin Films Containing Twin Boundaries"

_metals, doi:10.3390/met10070953_

Round 1

Reviewer 1 Report

This manuscript deals with molecular dynamics (MD) simulation of twin boundary (TB) effect on stress induced phase transition in iron thin films.  Although this work deals with the frequently studied topic, it brings some new results and conclusions.  The text of manuscript is clear and easy to understand. The paper can be published after improving some minority issues.

I think that references are incomplete. For example, after brief check I have found a very extensive study about influence of defects on alpha->epsilon transformation which is not mentioned: Luu et al. Metals  9, 1040 (2019).   

Authors should clearly declare Miller indices of studied TB in simulation method. In conclusions, authors discussed (111)bcc TBs in (3) but (112)bcc TB in (4), although the TB should be of the same type in both groups of films.

Authors should also discuss why dislocations was not found during the TB migration.

Author Response

Dear Reviewer,

many thanks for your comments, which are definitely helpful for us.

We answered your comments point by point and revised our manuscript. Please see the attatched file.

Hopefully the revised manuscript will be better than the previous version. Many thanks again for your work.

Best regards

                                                                                           Sincerely yours

                            Binjun, Wang; Yunqiang Jiang; Chun, Xu and Jianguo Zhang

Reviewer 2 Report

Everything in this paper is described properly, even though just a little work needs to be done. Variations of the parameters in the simulation is quite insufficient. Only a few things varied and simulating to get some figures. I have made some minor comments here which the authors need to address. 
  • Please add a nomenclature of the symbols used in the draft.
  • Please add more literature about pre-existing defects (not only dislocation), and focus the novelty of your work with them.
  • More attention should be paid to the application of punctuation marks; they are missing on many occasions.
  • Please avoid Reference Lumping and instead, locate them where they exactly belong.
  • Have you made any assumption designing the eight Fe thin films listed in table 1?
  • Stress-strain, phase transition, etc. are explained for film groups 1 and 2 separately. The authors are requested to make a conclusion that can clearly show the difference between those two groups. Conclusions are needed to be rearranged focusing the groups results patterns explained in the results section.

Author Response

(The authors gave the same response as above.)

Reviewer 3 Report

The authors conduct a molecular dynamic studies of thin films of Fe.
The main goal is a better understanding of the influence of different
ratios of twin boundaries on the strain induced structural phase transition from
bcc to a closed packed structure. While twin boundaries perpendicular to the
free surfaces do affect the critical strain for the structural phase transition,
the twin boundaries parallel to the surface affect it much less. In the latter
case, there are enough nucleation possibilities at the free surfaces.

The manuscript is written well and has a clear structure and aim.
Nevertheless there are few points I would like to see improved before
publication:

1. I would like to see more details about the creation of these twin boundaries.
Reproducibility is the highest value in research and I think there could be less
obstacles in order to continue numerical studies with well-known tools like
LAMMPS and ATOMSK. Please provide some more details or input examples for ATOMSK
in order to create your twin boundaries.
2. I think there are some few typos and inconsistencies in the text. Fig. 1c is
mentioned as film 7 in the caption but in the text as film 6. Please change
whatever is correct and check the text again carefully.
3. Following up on the last point, there are few word phrases which sound not
correct for me or could be improved, for example:
- keywords should have all the same style either uppercase or lowercase
- page 1: "the method nanoindentation", maybe only "using nanoindentation"?
- page 2: "dependency of free surface on the phase transition" I guess you want
exactly the opposite meaning?
- page 2: "the temperature changing induced phase transition"
- page 2: I think TB is not explained in the main text, only in the abstract.
- page 2: "decided factor" I understand the point but think that there is a
better formulation.
- page 12, line 314: "strain applying of film 5" We do apply strain to a film.
In general there could some sentences shortened in order to improve readability.
These are some examples but not all, please check the whole text carefully.
4. The figures are nice but I cannot agree on the "red" for unknown structures.
Either really use red or denote it was brown. By the way, green and red are
often not good because of people with dyschromatopsia. In addition, I am
wondering, if the subfigures (insets) in Figs 3, 8 and 12 are really needed.
They are not mentioned in the text and do not really show more than the main
plot, except for Fig. 3 maybe.
5. One point to the conclusion. Maybe there could be at the end one or two
general sentences about the main results of the whole manuscript?
6. The abstract could be improved or better cleaned up a bit. I would suggest to
adapt the order of sentences. The second sentences
could come later after describing the two sets, because you start with general
information about the films and then discuss the strain. Furthermore, I would
not shorten this sentence about the two film groups via "1(2)" but they could
be written out. Also in the following sentence "film group 1" and "film group 2"
might be really mentioned again the perpendicular and parallel.
This would improve the reading in my opinion.
7. Few minor points:
- there seems to be a discrepancy between author names and abbreviations in the
contributions section
- the x, y, z should be consistent, but e.g. in the caption of Table 1, the x is
uppercase.
- the same caption for Table 1, there is an additional period.

Author Response

(The authors gave the same response as above.)
